# Knowledge, Attitude, and Practice among Physical Therapists toward COVID-19 in the Kingdom of Saudi Arabia—A Cross-Sectional Study

**DOI:** 10.3390/healthcare10010105

**Published:** 2022-01-05

**Authors:** Adel Alshahrani, Ajay Prashad Gautam, Faisal Asiri, Irshad Ahmad, Mastour Saeed Alshahrani, Ravi Shankar Reddy, Mutasim D. Alharbi, Khalid Alkhathami, Hosam Alzahrani, Yasir S. Alshehri, Raee Alqhtani

**Affiliations:** 1Department of Medical Rehabilitation Sciences, College of Applied Medical Sciences, Najran University, Najran 11001, Saudi Arabia; amsalshahrani@nu.edu.sa (A.A.); rsalhyani@nu.edu.sa (R.A.); 2Department of Medical Rehabilitation Sciences, College of Applied Medical Sciences, King Khalid University, Abha 61413, Saudi Arabia; fasiri@kku.edu.sa (F.A.); iabdulhamed@kku.edu.sa (I.A.); msdalshahrani@kku.edu.sa (M.S.A.); rshankar@kku.edu.sa (R.S.R.); 3Department of Physical Therapy, Faculty of Medical Rehabilitation Sciences, King Abdulaziz University, Jeddah 21589, Saudi Arabia; malharbi@kau.edu.sa; 4Department of Health Rehabilitation, Shaqra University, Shaqra 11961, Saudi Arabia; kalkhthami@su.edu.sa; 5Department of Physical Therapy, College of Applied Medical Sciences, Taif University, Taif 21944, Saudi Arabia; halzahrani@tu.edu.sa; 6Department of Physical Therapy, College of Medical Rehabilitation Sciences, Taibah University, Madinah 41411, Saudi Arabia; yshehri@taibahu.edu.sa

**Keywords:** COVID-19, physical therapy, knowledge, attitude and practice

## Abstract

To curb the COVID-19 pandemic, the knowledge, attitude, and practice (KAP) of preventive measures play an essential role, and healthcare workers have had to endure a burden to care for COVID-19 patients. Thus, this study aimed to assess the weight of the KAP of physiotherapists in Saudi Arabia during the COVID-19 pandemic. This was a cross-sectional study, where we circulated an online KAP questionnaire to 1179 physical therapists, and among those, 287 participated and completed the questionnaire. The collected responses were analyzed using descriptive statistics, *t*-test, ANOVA, correlation, and regression analyses, and *p*-value ≤ 0.05 was considered statistically significant. Both males and females participated in almost equal numbers; most of the participants were <40 years, had a bachelor’s level of education, and were from the central region of Saudi Arabia. Social media and the internet were the primary sources of COVID-19-related information (74.6%). Knowledge components A (92%) and B (73.9%) were excellent among most participants. Approximately half of the participants (50.5%) had a moderate attitude toward COVID-19, and regarding the practice component, most participants (74.6%) scored moderately. Correlation analysis showed a low positive relationship between knowledge A, attitude, and practice components. Still, there was a very low positive relationship between knowledge B, attitude, and practice components, but both were statistically significant. Our study showed that physical therapists in Saudi Arabia exhibit good knowledge, attitude, and practice toward COVID-19.

## 1. Introduction

The COVID-19 pandemic started approximately two years ago, and the virus is transmitted through the air or droplets [1,2]. The most common symptoms associated with COVID-19 are fever, cough, headache, loss of taste or smell, and fatigue [2]. This pandemic affected the individual’s health and increased the burden on healthcare systems and the economy of almost all countries throughout the world. As a result, the World Health Organization (WHO) declared COVID-19 a global public health emergency [2]. In addition, international health bodies and national level agencies suggested several guidelines to control and prevent the spread of COVID-19 infections [2,3]. Unfortunately, approximately 254 million confirmed cases and 5.1 million deaths have already been reported [2].

The Kingdom of Saudi Arabia reported its first COVID-19 infection case in early March 2020 [4] and was one of the first countries to impose a nationwide lockdown to control the spread of infection [5]. The kingdom even stopped people’s entry into the country from all international borders, including land, water, and air [6]. Moreover, the Saudi Ministry of Health took several proactive steps, such as the mass screening of the population across the kingdom [7]. These measures adopted during the pandemic’s early stages have kept the illness under strict control. There was also a rapid decline in infections inside the country, and the recovery rate was very high with minimal mortality [8,9,10].

The physical therapy profession has been known to be one of the essential healthcare professions in improving patients’ health and function at all times. A physical therapist can help COVID-19 patients improve their respiratory functions, exercise endurance, and overall body strength [11,12]. However, since the pandemic began, physical therapy services ceased because COVID-19 can be transmitted from unrecognized COVID-19-infected patients to physical therapists. Similarly, asymptomatic therapists with COVID-19 may potentially infect their patients and cause significant health issues, especially in older people or patients with chronic diseases. As a result, all universities shifted their classes to online platforms for physical therapy students and used remote training for clinical work instead of in-person services due to closed clinics. However, this new approach in medical health education may limit students’ clinical experience in adhering to preventive measures, which predisposes students to the virus [12].

These appropriate government measures aim to protect physical therapists from exposure to the disease. However, awareness of COVID-19 is also contingent on precise information of the disease, a favorable attitude toward government efforts, and proper COVID-19 procedures. With the interruption of the outpatient and home rehabilitation service, it was necessary to reduce hospitalization rates and provide specific personalized telerehabilitation services, ensuring continuity of patient monitoring and improvement not only of the state of health but, above all, the quality of life. In this scenario, it became crucial to provide an efficient rehabilitation service and only through the appropriate knowledge, attitude, and practice of the therapists [13]. Good knowledge about the disease characteristics (knowledge A) and its transmission routes in addition to vulnerable groups (knowledge B) will positively impact attitude toward the disease, which, in turn, will transfer into clinical practice. All of these will prevent COVID-19 transmission to and from therapists and patients. There have been several studies conducted on the general public regarding knowledge, attitude, and practice toward COVID-19 in Saudi Arabia and globally [14,15,16,17,18,19,20,21,22,23,24,25,26,27,28,29,30,31]. However, no previous studies to our knowledge have assessed the knowledge, attitudes, and practice of physical therapists regarding the COVID-19 pandemic in Saudi Arabia. This will help us understand how many physical therapists, who are integral members of the team involved in COVID-19 patient’s care, know about the disease and what attitude they exhibit toward controlling it in their clinical practice. Therefore, the purpose of this study was to evaluate the knowledge, attitudes, and practice of physical therapists in Saudi Arabia toward COVID-19 during the pandemic.

## 2. Methodology

### 2.1. Design and Settings

This cross-sectional study was conducted between September 2020 and May 2021. The institutional Research Ethics Committee ethically approved the study proposal of Najran University (reference number- 25-05-07-20-EC). This study included physical therapists working in Saudi Arabia during the COVID-19 pandemic. The questionnaire was advertised and sent to participants through social media platforms as a link to a google form with details explaining the study, according to Strengthening the Reporting of Observational Studies in Epidemiology (STROBE) recommendations [32].

### 2.2. Outcome Measures

The participants were asked to fill the knowledge, attitude, and practice (KAP)-related questionnaire, which has already been used previously by Erfani et al. [15]. Before administering the questionnaire, it was sent to two expert researchers for their feedback. Moreover, ten physical therapists were invited for a pilot study to make sure that all the questions in the questionnaire were straightforward and easy to understand. Some minor edits and modifications were subsequently performed to some of the questions in the questionnaire. Finally, the researchers agreed on a 53-item questionnaire for data collection (Appendix A). In brief, the questionnaire included three sections: The first section was concerned with knowledge A and B, where knowledge A (questions K1–K18) was mainly focused on characteristics of the COVID-19 disease, and knowledge B (questions K19–K26) was more focused on its route of transmission and the groups at higher risk for the disease. The second section was about attitude (questions A1–15), concerning the opinions about detection, prevention, treatment, seriousness, and mortality of COVID-19, and the third section was about practice (questions P1–12), i.e., measures to follow for the prevention and control of COVID-19. Three reminders were sent to the participants to fill out their responses.

### 2.3. Statistical Analysis

All the analyses in this study were performed using the Statistical Package for Social Sciences software (version 26.0, IBM Corp., Armonk, NY, USA). Normality of data was tested using Kolmogorov–Smirnov test. Descriptive and inferential statistics involving *t*-test, ANOVA, correlation (Pearson product-moment analysis), and regression analysis were used to present results. Multiple regression analysis was also conducted to predict knowledge A, knowledge B, attitude, and practice component scores from different demographic variables. A *p*-value of less than or equal to 0.05 was considered statistically significant for each test.

## 3. Results

The questionnaire was sent to 1179 physical therapists under the Saudi Commission for Health Specialties (SCFHS). Among them, 416 agreed to participate, but only 311 filled the questionnaire responses, and, finally, 287 responses were considered for the final analysis. The questionnaire response rate of our study was 74.78%, which is regarded as a good response rate (>50% is deemed to be good) for online surveys. The flow of the study is shown below in Figure 1.

The results revealed 287 participants, with slightly more female participants than males (51.6% vs. 48.4%). Most of the respondents were Saudi residents working within the kingdom and also working in the physical therapy field, as shown in Table 1. The majority of participants were <40 years of age (94.4%) and had an educational level of bachelor’s degree (78.8%). Most of the respondents were from the central region (43.6%), followed by the western part (23.3%) and southern region (18.8%) of Saudi Arabia, as shown in Table 1.

Regarding the sources of information about COVID-19, the highest sources of information among the participants were social media and the internet, healthcare providers, and news media at a rate of 74.6% (214/287), 63.8% (183/287), and 53.3% (153/287), respectively, as shown in Table 2. However, there was a low degree of reliability to a source of information among the participants, with a range of scores (0–5) and a mean ± SD of 2.55 ± 1.202 (100%).

The study results showed that knowledge A toward COVID-19 was excellent in the majority of the participants (92%) with a total score, while knowledge B was also excellent among 73.9% of the participants. However, the participants’ attitude toward COVID-19 was almost moderate and poor at 50.5% and 38.7%, respectively, with a score of 38.79 ± 3.161. Furthermore, the highest degree of practice was moderate at 74.6%, with a score of 31.55 ± 2.878. A summary of these results is shown in Table 2.

The Pearson product-moment analysis was conducted to determine the correlations between knowledge A, knowledge B, attitude, and practice toward COVID-19. There was a reasonable positive correlation between knowledge A and knowledge B components, which was statistically significant (Table 3). The correlation between knowledge A and attitude components was low positive and statistically significant. Furthermore, the relationship between practice and attitude components was a low positive correlation and statistically significant. The correlations between knowledge B, attitude, and practice components were also statistically significant, but they had a very low correlation strength and, thus, were probably meaningless (Table 4).

For the age of participants and knowledge A, a statistically significant difference (one-way ANOVA) and post hoc analysis showed that between the age groups of 41–45, 46–50, 36–40, 50–60, and 31–35, there were statistically significant higher knowledge A scores of 73.1 ± 0.9, 72.6 ± 1.1, 72.1 ± 2.6, 71.7 ± 4.9 and 71 ± 3.5, respectively, compared to other age groups. A one-way ANOVA found a statistically significant difference between educational level and knowledge A. Post hoc analysis showed that participants with a doctoral level of education had a significantly higher score in knowledge A, with a score of 72.3 ± 2.1, compared to other groups. Additionally, there was a statistically significant difference between Saudi Arabia’s living regions and knowledge A (one-way ANOVA). Post hoc analysis determined a significantly higher knowledge A score of 70.9 ± 3.1 in the western region compared to other regions. Moreover, the results showed that the non-Saudi residents had significantly higher knowledge A scores than those of the Saudi residents. There was a statistically significant difference between the main work area and knowledge A score (one-way ANOVA), and post hoc analysis showed that participants from the government sectors had a significantly higher score in knowledge A of 72.2± 3.3 than those in other main work areas. The results also showed that those working in the non-physical therapy field had a significantly higher knowledge B score (72.38 ± 3.462) than those working in the physical therapy field. Moreover, there was a statistically significant difference between knowledge B and educational level (one-way ANOVA). Post hoc analysis showed that participants with a doctoral level of education had a significantly higher score of 23.1 ± 0.9 in knowledge B compared to other groups. The attitude component results showed a statistically significant difference between the regions of Saudi Arabia and attitude (one-way ANOVA), and post hoc analysis showed that the western region participants had a significantly higher score of 70.9 ± 3.1 in the attitude component compared to other regions. Moreover, there was a statistically significant difference between the main work area and attitude (one-way ANOVA). Post hoc analysis showed that participants from the government sectors had a significantly higher attitude score of 40.3 ± 2.4 than that of other main workgroups. These results are summarized in Table 5 and Table 6.

Regarding the sources of information about COVID-19, news media groups had a statistically significant knowledge A score, while scientific article/journal groups had statistically significant knowledge A and attitude scores (Table 7).

Multiple regression analysis was also conducted to predict knowledge A, knowledge B, attitude, and practice component scores for different demographic variables (Table 8). These variables statistically significantly predicted knowledge A scores (F (9, 277) = 5.199, *p* < 0.0001, R^2^ = 0.145, (14.5%)). Gender, age, and source of information variables added statistical significance to the prediction (*p* < 0.05). Moreover, these variables statistically significantly predicted knowledge B (F (1.974) = 5.199, *p* < 0.042, R^2^ = 0.060, (6%)). While these variables statistically significantly predicted attitude (F (9, 277) = 2.949, *p* < 0.002, R^2^ = 0.087, (8%)), the regions of Saudi Arabia and the main work variables added statistical significance to the prediction (*p* < 0.05). Finally, these variables could not statistically significantly predict the practice component.

## 4. Discussion

Several components of human behavior will decide the fate of this COVID-19 pandemic. Among those components, the most important ones are knowledge about the disease, attitude toward the disease, and practices people follow to check or control the disease [15]. The KAP of healthcare providers, including physical therapists, toward the COVID-19 disease will play an important role in controlling the pandemic. The findings of our study reveal greater COVID-19 KAP scores overall in female participants and those with a higher educational level and professional groups. Furthermore, these scores were higher in certain age groups.

Several studies have been conducted worldwide, including in Saudi Arabia, about the knowledge, attitude, and practice (KAP) among general and chronic illness populations for the COVID-19 disease [15,16,17,18,19,20,21,22,23,24,25]. Few studies were also conducted in Saudi Arabia on healthcare professionals and students [26,27,28,29,30]. Being an integral part of the COVID-19 patients’ rehabilitation team, physical therapists are at very high risk for COVID-19 infections [34,35,36,37,38]. Physical therapists must have detailed information about COVID-19 and should follow the standard guidelines set by regulatory authorities while handling the patients during their clinical practice throughout this ongoing pandemic.

### 4.1. Knowledge Component

The knowledge of the participants was very good regarding COVID-19. There was no significant difference for the knowledge component among gender, those working or not working in the physical therapy field, and those studying or practicing in Saudi Arabia. However, there were significant differences among different age groups for the knowledge A component due to the large range of participants’ ages. In addition, the participants’ educational background, nationality (Saudi vs. non-Saudi), regional location, and work status showed significant differences for the knowledge A component. Still, there was no significant difference among participants for the knowledge B component, except for the educational background. Our study showed almost similar trends as those reported by previous studies conducted in Saudi Arabia or other countries [14,15,17,18,19,20,22,23,24,26,27]. The researchers found the same trends in applied medical sciences [31] and dental science professionals [28] in Saudi Arabia.

Previous studies had comparatively less knowledge about COVID-19 disease transmission than that in our study [15,16,17,18,19,20,21,22,23,24,25,26,27]. This may be due to the involvement of the general population in these studies when compared to our study’s population, which was healthcare professionals. Another reason for the high knowledge scores was mainly due to the awareness information available on social media platforms and the information provided by the Saudi government Ministry of Health media portals and mass media from the beginning of the COVID-19 outbreak. The western region’s physical therapists showed better scores in the knowledge component. This may be due to this region being the oldest and having the most advanced cities with more awareness campaigns and stricter norms regarding COVID-19 control. Moreover, as our participants are graduate level and above in the healthcare profession, they are bound to have more knowledge about the disease. There was a positive correlation between the knowledge of COVID-19 and the participant’s educational status, which was similar to that reported in other international studies [1,2,4,7,8,10,11,12,13,16,17,18,21,25,26,28,30,32,33,34,35,36,37,38]. In addition to mass media, our participants also reported that scientific journals were a major source of information about COVID-19.

### 4.2. Attitude Component

Our study’s participants showed a positive attitude toward COVID-19, and there were no significant differences among the gender, nationality, and education status of the participants for COVID-19. However, the main work area and the region where the participant lives in Saudi Arabia showed a significant difference in attitude scores. The participants were aware of the virus transmission rates, methods to prevent the transmission, and the various recommendations to isolate virus-infected patients. Questionnaire responses were above 90% for lockdown implementation, quarantine facilities, travel restrictions, and shutting down the educational facilities and prayer centers temporarily to curb the pandemic spread. These responses may be due to the previous experience gained with SARS/MERS infection spread in the western regions of Asia [39,40,41,42,43]. The majority of the participants (61.3%) exhibited a positive attitude toward the best practice guidelines for hand hygiene, and only 38.7% showed a poor attitude toward this practice in our study. Our scores were better than those of a previously conducted study, as, in that study, all the participants were students [31], whereas, in our study, we included physical therapists who were already practicing. The poor attitude by around 39% of the participants is not a good sign, and this gap needs to be addressed by increasing health education toward ways to prevent pandemic spread. Moreover, good knowledge scores correlated well with the attitude toward preventing the virus spread. This was similar to the trends shown in previous studies [8,20,28,30,36].

### 4.3. Practice Component

Our study participants showed that they follow the best practice behavior toward preventing COVID-19. There were no significant differences among gender, region, nationality, and education status for self-prevention practices against COVID-19. Overall, the practice rating scores were good, with most of the participants (85%) exhibiting the best practices to prevent themselves from becoming infected by COVID-19. There was a good correlation between knowledge and attitude toward best practices. This indicates that individuals with good knowledge and attitude toward the prevention of COVID-19 will likely follow the best practice. Our study results showed similar results to those of previous studies in healthcare professionals [8,20,28,30,36,44]. Our study participants followed practice guidelines better than previously conducted studies, which were conducted on the general public [4,7,8,11,13,18,21,26,30,38,39]. This may be because our participants are in the healthcare profession and are more likely better at understanding the need to follow best practices to prevent themselves from COVID-19 infections than the general public. These best practice responses are also attributed to the day-by-day updated guidelines provided by the health authorities in Saudi Arabia and the World Health Organization through mass media and social networking platforms [6,22,41].

Overall, our study showed good knowledge among physical therapy professionals in Saudi Arabia about COVID-19 infections and their prevention. Moreover, physical therapists exhibited a good attitude and practice toward dealing with the prevention of COVID-19 spread. These results show the positive impacts of the widespread information available across the internet and social media platforms and the Saudi government’s and WHO’s efforts to provide information about the COVID-19 pandemic and its control measures. We are aware that the present study is not free from limitations. First, a regression approach would require a larger sample. Second, the study was unable to distinguish pre-existing KAP skills from newly acquired ones. Third, it would have been useful to use widespread, standardized, and validated scales despite the fact that a panel of experts had formulated the online questionnaire, which should be reliable for all physiotherapists involved. Thus, future research is needed to address the above limitations.

## 5. Conclusions

Our study showed that physical therapists in Saudi Arabia exhibit good knowledge, attitude, and practice toward COVID-19. This indicates that they are essentially responsible members of the COVID-19 care team and have to implement these in their clinical practice. This also shows the success of the government of Saudi Arabia and the Ministry of Health in curbing the COVID-19 pandemic in the kingdom.

## Figures and Tables

**Figure 1 healthcare-10-00105-f001:**
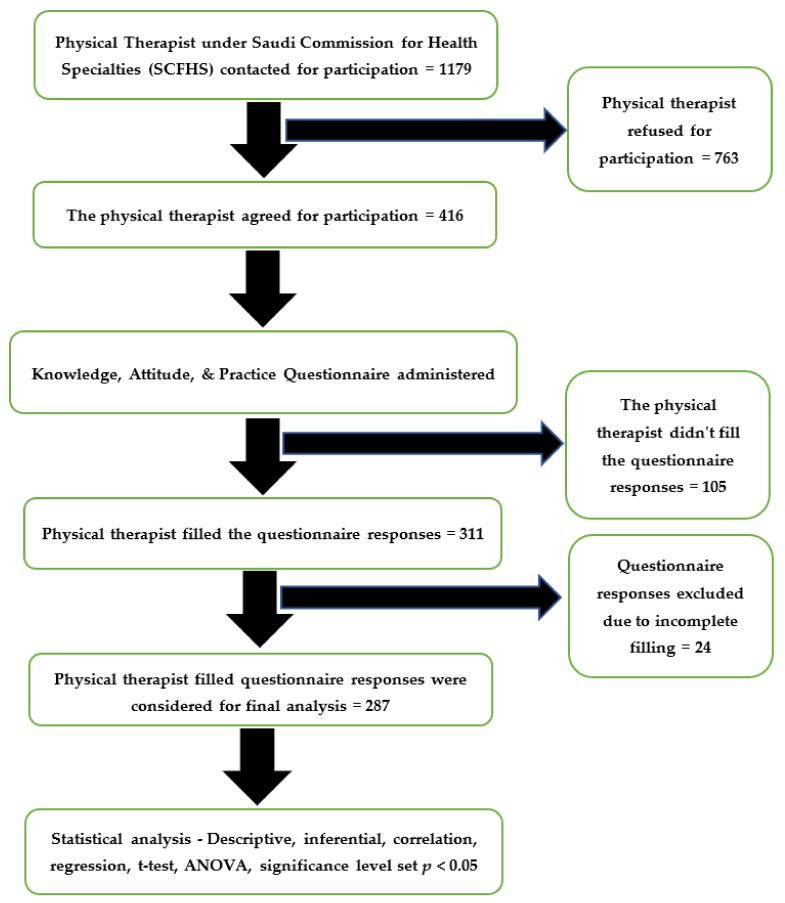
The flow of participants throughout the study.

**Table 1 healthcare-10-00105-t001:** Descriptive characteristics of participants (*N* = 287).

	Characteristics	Number (%)
1	Age Group	
	≤40 years	271 (94.42)
	>40 years	16 (5.58)
2	Gender	
	Male 29 ± 13.5	139 (48.43)
	Female 26 ± 8.7	148 (51.57)
3	Nationality	
	Saudi	269 (93.73)
	Non-Saudi	18 (6.27)
4	Education Level	
	Diploma	8 (2.79)
	Bachelor’s degree	226 (78.74)
	Master’s Degree	31 (10.80)
	Doctoral Degree	22 (7.67)
5	Living Region in Saudi Arabia	
	Eastern	27 (9.40)
	Western	67 (23.34)
	Central	125 (43.55)
	Northern	14 (4.87)
	Southern	54 (18.81)
6	Main Work Area	
	Healthcare Clinic	98 (34.14)
	Ministry of Health	69 (24.04)
	Academic	44 (15.33)
	Private hospital	45 (15.67)
	Other government sectors	31 (10.80)
7	Practicing in Saudi Arabia	
	YES	281 (97.90)
	NO	6 (2.09)

**Table 2 healthcare-10-00105-t002:** Sources of information for COVID 19 of the participants (*N* = 287).

Source of Information	Yes	No
	Number (%)	Number (%)
News media	153 (53.3)	134 (46.7)
Social media and internet	214 (74.6)	73 (25.4)
Family/friends	67 (23.3)	220 (76.7)
Scientific articles/journals	114 (39.7)	173 (60.3)
Healthcare providers	183 (63.8)	104 (36.2)

**Table 3 healthcare-10-00105-t003:** Number of questions, range, total scores, and levels of knowledge, attitude, and practice regarding COVID-19.

Variables	Number of Questions	Range of Score	Total Score (Mean ± SD)	Level (%), *N* = 287		
				Poor	Moderate	Excellent
Knowledge A	18	18–54	49.96 ± 3.79	0 (0%)	23 (8%)	264 (92%)
Knowledge B	8	0–8	42.18 ±1.51	7 (2.4%)	68 (23.7%)	212 (73.9%)
Attitude	15	15–45	38.79 ±3.16	111 (38.7%)	145 (50.5%)	31 (10.8%)
Practice	12	12–36	31.55 ± 2.88	43 (15%)	214 (74.6%)	30 (10.5%)

**Table 4 healthcare-10-00105-t004:** Correlations between KAP components.

Variables	r	*p*-Value
Knowledge A, knowledge B	0.433 **	0.0001
Knowledge A, attitude	0.253 **	0.0001
Knowledge A, practice	0.157 **	0.008
Knowledge B, attitude	0.205 **	0.0001
Knowledge B, practice	0.146 **	0.013
Practice, attitude	0.37 **	0.0001

r = 0–0.2: very low and probably meaningless. r = 0.2–0.4: a low correlation that might warrant further investigation. r = 0.4–0.6: a reasonable correlation. r = 0.6–0.8: a high correlation. r = 0.8–1.0: a very high correlation (reference range) [33]. ** Significant correlation if *p* value ≤ 0.05

**Table 5 healthcare-10-00105-t005:** Variance in demographic characteristics and the knowledge test A, knowledge test B, attitude, and practice scores of COVID-19 (*N* = 287).

Variables	Number	Knowledge A			Knowledge B			Attitude			Practice		
		Mean ± SD	t/F *	*p* **	Mean ± SD	t/F *	*p* **	Mean ± SD	t/F *	*p* **	Mean ± SD	t/F *	*p* **
Are you working in the physical therapy field?													
No	8	72.38 ± 3.462	1.998	0.083	23.38 ± 0.916	3.662	0.006 **	39.2 ± 2.2	0.581	0.577	31.7 ± 3.4	0.169	0.870
Yes	279	69.89 ± 3.77			22.14 ± 1.510			38.7 ± 3.1			31.5 ± 2.8		
Are you practicing in Saudi Arabia?													
No	8	70.1 ± 4.3	0.111	0.915	22.6 ± 1.5	0.804	0.446	37.8 ± 3.7	−0.699	0.506	30.2 ± 2.8	1.294	0.233
Yes	279	69.9 ± 3.7			22.1 ± 1.5			38.8 ± 3.1			31.5 ± 2.8		
Gender													
Male	139	69.7 ± 4.3	0.964	0.336	22.1 ± 1.7	0.054	0.957	38.7 ± 3.3	352	0.725	31.4 ± 2.9	0.616	0.539
Female	148	70.1 ± 3.1			22.1 ± 1.2			38.8 ± 3			31.6 ± 2.8		
Age													
<20	2	66 ± 4.2	3.766	0.001 **	21.5 ± 0.7	1.352	0.226	33.5 ± 0.7	10.877	0.073	31 ± 0	0.761	0.621
20–30	215	69.4 ± 3.7			22 ± 1.5			38.5 ± 3.2			31.5 ± 2.9		
31–35	37	71 ± 3.5			22.4 ± 1.3			39.5 ± 2.5			31.5 ± 2.7		
36–40	19	72.1 ± 2.6			22.6 ± 0.8			39.3 ± 2.9			31.8 ± 2.9		
41-45	6	73.1 ± 0.9			22.5 ± 0.8			40.1 ± 2			31.8 ± 2.3		
46–50	3	72.6 ± 1.1			23.3± 1.1			41.3 ± 1.5			30± 2		
50–60	4	71.7 ± 4.9			22.5± 1.7			38 ± 4.2			32.5 ± 1.9		
>61	1	76 ± 0			24 ± 0			39 ± 0			26 ± 0		

* Fisher exact test and *t*-test. ** Significant correlation if *p* value ≤ 0.05.

**Table 6 healthcare-10-00105-t006:** Variance of demographic characteristics and the knowledge test A, knowledge test B, attitude, and practice scores of COVID-19 (*N* = 287).

Variables	*N*	Knowledge A			Knowledge B			Attitude			Practice		
		Mean ± SD	t/F *	*p* **	Mean ± SD	t/F *	*p* **	Mean ± SD	t/F *	*p* **	Mean ± SD	t/F *	*p* **
What is your educational background?													
Diploma	8	71.5 ± 1	4.566	0.001 **	22.5 ± 1	2.650	0.034 **	39 ± 1.6	1.978	0.098	32.5 ± 1.9	0.295	0.881
Bachelor	226	69.9 ± 3.6			22.2 ± 1.2			39.1 ± 2.8			31.5 ± 2.7		
Master	31	70.7 ± 3.8			22.1 ± 1.7			38.9 ± 2.8			31.1 ± 3.1		
Doctoral	22	72.3 ± 2.1			23.1 ± 0.9			39.5 ± 2.6			31.7 ± 2.5		
Nationality													
Saudi	269	69.8 ± 3.8	2.031	0.05 **	22.1 ± 1.5	1.154	0.261	38.7 ± 3.1	0.330	0.745	31.5 ± 2.9	0.363	0.720
Non-Saudi	18	71.3 ± 3			22.5 ± 1.2			39.1 ± 3.5			31.3 ± 1.8		
In which part of Saudi Arabia do you live?													
Central	125	69.9 ± 3.9	3.766	0.005 **	22.2 ± 1.4	1.581	0.179	39.1 ± 3.1	5.029	0.001 **	31.4 ± 2.7	0.728	0.573
Eastern	27	70.7 ± 2.8			22.1 ± 1.1			39.1 ± 2.5			31.2 ± 2.3		
Western	67	70.9 ± 3.1			22.3 ± 1.2			39.3 ± 2.6			32 ± 2.5		
Northern	14	70.2 ± 1.8			21.7 ± 1.4			38.2 ± 3.7			31.2 ± 3.1		
Southern	54	68.3 ± 4.3			21.8 ± 1.9			37.1 ± 3.4			31.3 ± 3.6		
Which of the following is describes your main work?													
Health Clinics	100	69.2 ± 3.8	3.950	0.004 **	22 ± 1.7	0.779	0.539	38.1 ± 3.3	3.357	0.011 **	31.8 ± 2.9	0.811	0.519
Ministry of health	67	70.1 ± 3.5			22.2 ± 1.2			38.7 ± 2.9			31.4 ± 2.8		
Academic university	44	70 ± 4.1			22.1 ± 1.6			38.6 ± 3.6			31 ± 3.2		
Private Hospital	45	69.7 ± 3.1			22.1 ± 1.1			39.4 ± 2.6			31.8 ± 2.4		
Government sectors	31	72.2 ± 3.3			22.5 ± 1.1			40.3 ± 2.4			31.2 ± 2.6		

* Fisher exact test and *t*-test. ** Significant correlation if *p* value ≤ 0.05.

**Table 7 healthcare-10-00105-t007:** Source of information among the participants and its correlation with knowledge, attitude, and practice.

Variables	*N*	Knowledge A			Knowledge B			Attitude			Practice		
		Mean ± SD	t *	*p* **	Mean ± SD	t *	*p* **	Mean ± SD	t *	*p* **	Mean ± SD	t *	*p* **
News media													
No	134	69.4 ± 4.1	1.963	0.05 **	22.1 ± 1.7	0.906	0.366	38.6 ± 3.4	0.796	0.427	31.3 ± 3.2	1.226	0.221
Yes	153	70.3 ± 3.3			22.2 ± 1.2			38.9 ± 2.9			31.7 ± 2.5		
Social media and the internet													
No	73	69.5 ± 4.7	0.849	0.398	21.9 ± 2.1	1.214	0.228	38.4 ± 3.5	−0.887	0.377	31.4 ± 2.9	0.409	0.683
Yes	214	70.1 ± 3.3			22.2 ± 1.2			38.8 ± 3			31.5 ± 2.8		
Family/friends													
No	220	70 ± 3.7	0.331	0.741	22.2 ± 1.5	0.456	0.650	38.9 ± 3.1	1.606	0.111	31.5 ± 2.8	0.222	0.825
Yes	67	69.8 ± 3.9			22.1 ± 1.4			38.2 ± 3.4			31.4 ± 2.9		
Scientific articles/journals													
No	173	69.2 ± 3.8	3.809	0.0001 **	22.1 ± 1.6	1.741	083	38.4 ± 3.3	2.022	0.044 **	31.6 ± 3	0.488	0.626
Yes	114	70.9 ± 3.5			22.3 ± 1.2			39.2 ± 2.8			31.4 ± 2.6		
Healthcare providers													
No	104	68.8 ± 4	3.534	0.001 **	22 ± 1.6	0.810	0.419	38.1 ± 3.6	2.589	0.010 **	31.3 ± 3.3	0.629	0.530
Yes	183	70.5 ± 3.5			22.2 ± 1.4			39.1 ± 2.7			31.6 ± 2.5		

* *t*-test. ** Significant correlation if *p* value ≤ 0.05.

**Table 8 healthcare-10-00105-t008:** Multiple linear regression of association between participants’ demographic characteristics with knowledge A, knowledge B, attitude, and practice scores of COVID-19.

Variables	Knowledge A			Knowledge B			Attitude			Practice		
	Unstandardized Coefficient (B)	*p* **	VIF *	Unstandardized Coefficient (B)	*p* **	VIF *	Unstandardized Coefficient (B)	*p* **	VIF *	Unstandardized Coefficient (B)	*p* **	VIF *
Working in the physical therapy field	−0.950	0.498	1.204	−1.081	0.066	1.204	−0.376	0.756	1.204	−0.407	0.722	1.204
Working or practicing in Saudi Arabia	0.523	0.699	1.125	−0.247	0.663	1.125	1.119	0.339	1.125	1.379	0.214	1.125
Gender	0.974	0.031 **	1.143	0.098	0.602	1.143	0.266	0.493	1.143	0.130	0.724	1.143
Age	0.631	0.018 **	1.646	0.065	0.560	1.646	0.101	0.661	1.646	−0.073	0.737	1.646
Educational level	0.262	0.258	1.801	0.153	0.116	1.801	0.134	0.502	1.801	0.041	0.828	1.801
Nationality	0.854	0.373	1.223	0.191	0.633	1.223	0.439	0.595	1.223	0.204	0.794	1.223
Part of Saudi Arabia	−0.128	0.378	1.111	−0.101	0.095	1.111	−0.336	0.007 **	1.111	−0.022	0.855	1.111
Main work	0.288	0.112	1.414	−0.010	0.896	1.414	0.319	0.042 **	1.414	−0.096	0.518	1.414
Source of information	0.518	0.005 **	1.089	0.062	0.419	1.089	0.206	0.191	1.089	0.085	0.572	1.089

* Variance inflation factor (VIF). ** Statistically significant at *p* < 0.05.

## Data Availability

On request to the corresponding author Ajay Prashad Gautam (agautam@kku.edu.sa), all data are available at the Department of Medical Rehabilitation Sciences.

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
