# Peer review of "Knowledge, Attitude, and Practice among Physical Therapists toward COVID-19 in the Kingdom of Saudi Arabia—A Cross-Sectional Study"

_healthcare, 2022, doi:10.3390/healthcare10010105_

Round 1

Reviewer 1 Report

Thank you for the opportunity to review the manuscript. The purpose of the study is clearly detailed and the methodology appropriate for this purpose. 

Title:

The title of this study is misleading as clinical decision making is not discussed in the background section or addressed through the survey. 

Background:

The authors need to expand on how knowledge, practice, and attitude extends or translates to patient care or the daily practice of physical therapists. 

While the research purpose is clear, it is not clear from this section why this is a relevant question and of interest to the readers of this journal. 

Methods:

The authors detail out the response to the survey but need to calculate the response rate. What is an acceptable survey response rate? Was this an adequate response rate to answer the research question?

How many times was the survey distributed? 

The readers would benefit from learning the demographic information about the physical therapist population in Saudi Arabia. How representative is the study sample? 

Figure 3: Need to elaborate on the abbreviation "MOH"

The survey is not included in the Appendix even though the authors indicate that it is.

Knowledge A and B need to be defined. 

Figure 4 is labeled as "Source of information about general population." This is an inaccurate label about a figure that is about the study population - a specific group of healthcare providers. 

Discussion:

The authors need to include a citation for the first two lines of the discussion. 

The second paragraph is a repeat of the results section and the first three sentences can be deleted. 

The authors found that Knowledge about this particular phenomenon is higher in physical therapists from the Western region of the country but do not expand on why this may be the case. 

The authors make the claim that the results of this study speak to the positive efforts of the government and the World Health Organization. This conclusion cannot be drawn from this study findings. Figure 4 indicates that the primary source of information is social media. 

What are the implications of this study's findings? 

Conclusion

General comments: 

Require English editing throughout. For example, in the Discussion section under Practice - the first sentence of the paragraph is incomplete. The second sentence contains the word "regional" which should most likely be "region."

There is a sentence in the discussion section that is all in upper case - it is unclear why this is in upper case. 

Author Response

Response to Reviewer comments

Thank you for your effort and time in reviewing our manuscript. The reviewing process has significantly improved the quality of this manuscript. Therefore, I am submitting this "Response to reviewers" document summarizing the changes we made in response to the critiques.

Reviewer 1

S.N.

Queries

Response to queries

Line no.

1

Title: The title of this study is misleading as clinical decision making is not discussed in the background section or addressed through the survey. 

We have added the clinical decision making as suggested

3-4

2

The authors need to expand on how knowledge, practice, and attitude extends or translates to patient care or the daily practice of physical therapists. 

We have added the information as suggested

73-77

3

While the research purpose is clear, it is not clear from this section why this is a relevant question and of interest to the readers of this journal. 

We have added the information as suggested

81-84

4

The authors detail out the response to the survey but need to calculate the response rate. What is an acceptable survey response rate? Was this an adequate response rate to answer the research question?

We have added the information as suggested

125-126

5

How many times was the survey distributed?

We have added the info as suggested

111

6

The readers would benefit from learning the demographic information about the physical therapist population in Saudi Arabia. How representative is the study sample? 

We have added the information as suggested

Table 1

7

Figure 3: Need to elaborate on the abbreviation "MOH"

We have replaced figures 2 & 3 with Table 1

Table 1

8

The survey is not included in the Appendix even though the authors indicate that it is.

We have added the information as suggested

Appendix

9

Knowledge A and B need to be defined. 

We have added the info as suggested

103-111

10

Figure 4 is labeled as "Source of information about general population." This is an inaccurate label about a figure that is about the study population - a specific group of healthcare providers. 

Figure 4 has become Figure 2 now, and the label has been corrected as suggested

Figure 2

186

11

Discussion:

The authors need to include a citation for the first two lines of the discussion. 

The second paragraph is a repeat of the results section and the first three sentences can be deleted. 

The authors found that Knowledge about this particular phenomenon is higher in physical therapists from the Western region of the country but do not expand on why this may be the case. 

The authors make the claim that the results of this study speak to the positive efforts of the government and the World Health Organization. This conclusion cannot be drawn from this study findings. Figure 4 indicates that the primary source of information is social media. 

We have added  and corrected the information as suggested

Added as suggested

Deleted as suggested

Added as suggested

Added and corrected as suggested

274

303-309

350-351

355-356

12

What are the implications of this study's findings? 

We have added the information as suggested

363-364

13

General comments: 

Require English editing throughout. For example, in the Discussion section under Practice - the first sentence of the paragraph is incomplete. The second sentence contains the word "regional" which should most likely be "region."

There is a sentence in the discussion section that is all in upper case - it is unclear why this is in upper case. 

We have corrected the information as suggested

We have corrected the information as suggested

336-337

300

Reviewer 2 Report

The manuscript can have a general merit especially in a journal such as Healthcare, but to be suitable for publication it requires a massive revision of the methods (clarity on the questionnaire and statistical tools) and above all in the description of the results. Curiously, I found a nice evaluation of the results in discussion, but which I did not find clear in the respective section.

Re-enter the line numbers

The abstract is unstructured. In the background the intervention rationale is missing, then a solid methods section, after a draft of the objective there is immediately the result of the investigation ...

The figure 1 needs to be move in results section..

From methods section I cannot understand the KAP A and the KAP B, from the ref I retrieved this sentence but its misleading.. “The questions regarding the knowledge of the participants about COVID-19 was divided into two sections, one regarding their knowledge about the characteristics of the disease (Knowledge Test A) and the other regarding what they knew about the transmission routes and groups at higher risk for the disease (Knowledge Test B).”
In addition to explaining it, can I recommend a table or figure to justify the division? it is crucial to explain the correlations in the results.

In statistical analysis correlation method and software calculation are missing

Figure 2,3 and 4 are made with Excel suite… I find it a bit mortifying for the big effort, I recommend when plotting a barplot to insert standard deviation bars. So, I suggest a table with the characteristics of the respondents to the questionnaire to get an overview of the sample

Please explain the tables… Why not use logistic regression for dichotomous questions? Why did you use Fisher if they are parametric distributions? Why did you use the Fisher? unexplained abbreviations(VIF?), there is a capital P with two asterisks at the top, but then in the column it is not clear what is significant or not. Please explain the Multiple linear regression in methods ..

“””” The participants were from almost all the regions of Saudi Arabia, but the majority were from the country's central region. The findings of our study reveal significantly greater KAP overall scores about COVID-19 KAP in female participants and higher educational level and professionals' groups. Furthermore, these scores were gradually increased with advancing age  “””
They are more clear findings but which I do not find well outlined in the results section.

Limitation section totally missing

Author Response

Response to Reviewer comments

Thank you for your effort and time in reviewing our manuscript. The reviewing process has significantly improved the quality of this manuscript. Therefore, I am submitting this "Response to reviewers" document summarizing the changes we made in response to the critiques.

Reviewer 2

S.N.

Queries

Response to queries

Line no.

1

The abstract is unstructured. In the background the intervention rationale is missing, then a solid methods section, after a draft of the objective there is immediately the result of the investigation ...

As per journal requirements abstract should be unstructured. We have added the extra information as suggested

19-23

2

The figure 1 needs to be move in results section..

We have shifted as suggested

122-166

3

From methods section I cannot understand the KAP A and the KAP B, from the ref I retrieved this sentence but its misleading.. “The questions regarding the knowledge of the participants about COVID-19 was divided into two sections, one regarding their knowledge about the characteristics of the disease (Knowledge Test A) and the other regarding what they knew about the transmission routes and groups at higher risk for the disease (Knowledge Test B).”
In addition to explaining it, can I recommend a table or figure to justify the division? it is crucial to explain the correlations in the results.

We have added the information as suggested in outcome section and also added the questionnaire in the appendix

Correlations between KAP components has already been explained previously in results section as table 2 now it’s available in table 3

103-111

213

Table 3

4

Figure 2,3 and 4 are made with Excel suite… I find it a bit mortifying for the big effort, I recommend when plotting a barplot to insert standard deviation bars. So, I suggest a table with the characteristics of the respondents to the questionnaire to get an overview of the sample

We have replaced the figures 2 & 3 with Table 1

Table 1

5

Please explain the tables… Why not use logistic regression for dichotomous questions? Why did you use Fisher if they are parametric distributions? Why did you use the Fisher?

unexplained abbreviations(VIF?),

there is a capital P with two asterisks at the top, but then in the column it is not clear what is significant or not.

Please explain the Multiple linear regression in methods

We have used fisher test as we were having only few observations, so this will be a better test than logistic regression

VIF* is the variance inflation factor, and this information was already there in below the respective table 7

P** information was already there in below the tables P<0.05 was considered statistically significant now we have added ** inside the table for significant findings

About Multiple linear regression, we have added the information in the statistical analysis section as suggested

269

Table 4-7

116-119

6

“””” The participants were from almost all the regions of Saudi Arabia, but the majority were from the country's central region. The findings of our study reveal significantly greater KAP overall scores about COVID-19 KAP in female participants and higher educational level and professionals' groups. Furthermore, these scores were gradually increased with advancing age  “””
They are more clear findings but which I do not find well outlined in the results section.

We have corrected the information as suggested

284-286

7

Limitation section totally missing

We have added the information as suggested

357-360

Round 2

Reviewer 2 Report

Title: Cross Sectional

18-20: To curb the COVID-19 pandemic, knowledge, attitude and practice (KAP) of preventative measures play an essential role and healthcare workers have had to endure a burden to care for COVID-19 patients. Thus, this study aims to assess the weight of KAPs of physiotherapists in Saudi Arabia during the covid-19 pandemic.

70 : I suggest a statement like this with ref: “With the interruption of the outpatient and home rehabilitation service, it was necessary to reduce hospitalization rates, provide specific personalized telerehabilitation services, ensuring continuity of patient monitoring and improvement not only of the state of health but above all of the quality of life. In this scenario, it became crucial to provide an efficient rehabilitation service and only through the appropriate Knowledge, Attitude, and Practice of the therapists.” (ref: https://doi.org/10.1108/JET-11-2020-0047)

120 great

Figure 2. Without the standard deviation bars, better to remove .. a graph built on excel makes the manuscript lose authority

268 The first sentence of the discussion must paraphrase the objective of the study, subsequently the major findings of the study must be highlighted

268-281 These are all statements already made explicit in the introduction

355 I suggest enriching the limitations with statements like these: “We are aware that the present study is not free from limitations. First, a regression approach would require a larger sample. Second, the study was unable to distinguish pre-existing KAP skills from newly acquired ones. Third, it would have been useful to use widespread, standardized, and validated scales, despite that a panel of experts had formulated the online questionnaire that should be reliable for all physiotherapists involved. Thus, future research is needed to address the above limitations.” (ref:  https://doi.org/10.3390/ijerph18189676 )

Author Response

We would like to convey our sincere thanks for giving your valuable suggestions for improvement of our study mauscript. We have tried our level best for answering all your queries.

Thanks again and wishing you all A very happy and prosperous new year 2022 form our team
